# Abnormal Alpha-1 Antitrypsin Levels and Other Risk Factors Associated with Lung Function Impairment at 6 and 12 Months after Hospitalization Due to COVID-19: A Cohort Study

**DOI:** 10.3390/healthcare10122341

**Published:** 2022-11-22

**Authors:** Beatriz María Jiménez-Rodríguez, Eva Maria Triviño-Ibáñez, José Gutiérrez-Fernández, Ana Dolores Romero-Ortiz, Eldis Maria Ramos-Urbina, Concepción Morales-García

**Affiliations:** 1Department of Pneumology, University Hospital Virgen de las Nieves, 18014 Granada, Spain; 2Department of Microbiology, School of Medicine and PhD Program in Clinical Medicine and Public Health, University of Granada-IBS, 18010 Granada, Spain; 3Biomedical Research Institute of Granada-IBS, 18012 Granada, Spain; 4Department of Nuclear Medicine, Hospital Universitario Virgen de las Nieves, 18014 Granada, Spain; 5Department of Microbiology, University Hospital Virgen de las Nieves, 18014 Granada, Spain

**Keywords:** COVID-19, respiratory function tests, alpha-1 antitrypsin, long-term follow-up

## Abstract

Respiratory function deficits are common sequelae for COVID-19. In this study, we aimed to identify the medical conditions that may influence lung function impairment at 12 months after SARS-CoV2 infection and to analyze the role of alpha-1 antytripsin (AAT) deficiciency (AATD). A cohort study was conducted on hospitalized COVID-19 pneumonia patients in Granada (Spain) during the first infection wave who were referred to a post-COVID-19 hospital clinic. The patients were monitored with three follow-up visits from May 2020 to May 2021. Previous medical history, hospital admission data, baseline parameters and physical examination data were collected at the first visit. Pulmonary function tests were performed at 6 and 12 months together with the determination of AAT level and AATD genotype. After 12 months, 49 out of 157 patients (31.2%) continued to have lung function impairment. A multivariate analysis showed a statistically significant association of lung function impairment with: higher Charlson index; pneumonia with a central and/or mixed distribution; anemia on admission; time in intensive care; need for corticosteroid boluses; abnormal respiratory sounds at 6 months; elevated lactate dehydrogenase at 12 months; abnormal AAT; and MZ genotype. Our results suggest that these medical conditions predispose COVID-19 patients to develop long-term lung function sequelae.

## 1. Introduction

The term “Post-COVID-19 conditions” defined by the World Health Organization (WHO) and the Center for Disease Control and Prevention (CDC) is currently used to refer to continuous or new persistent symptoms, secondary sequelae from organ damage and effects of the disease or hospitalization that appear after the acute infection caused by Severe Acute Respiratory Syndrome (SARS) coronavirus 2 (SARS-CoV-2) [1,2]. Since other systemic viral diseases [3] and epidemics caused by previous coronaviruses (e.g., SARS-CoV-1, Middle East Respiratory Syndrome coronavirus [MERS]) have been associated with post-infectious sequelae and long-term complications [4,5], it was expected that they would also develop in post-COVID-19 patients. In this sense, alterations in respiratory function secondary to COVID-19 infection have been described, but they are not always related to the severity of the disease [6,7].

The CDC proposed α1-antitrypsin (AAT) deficiency (AATD) as a possible medical condition deserving further study due to mixed levels of evidence [8]. AAT is a glycoprotein belonging to the serpin group whose main function is to inhibit neutrophil elastase and prevent excessive proteolytic degradation of the connective tissue of the lungs [9,10]. AATD is a rare genetic disease due to mutations of the *SERPINA1* gene that produce low levels or defective AAT in the blood. This increases the risk of developing a variety of diseases, including pulmonary emphysema. Recent studies have described how this protein has biological functions that can counteract both SARS-CoV-2 infection and the underlying pathophysiological processes. It has also been suggested that patients with genotypic alterations have a higher risk of severity and even death [11,12]. In addition, a 2004 analysis of previous epidemics with other coronaviruses showed that serum samples from SARS patients had dramatically elevated levels of truncated forms of AAT, which correlated with the severity of SARS and indicated that these truncated forms of AAT could serve as SARS biomarkers with 100% sensitivity [13].

With the above in mind, the primary objective of the present study was to identify which medical conditions may influence the development of impaired lung function 12 month after COVID-19 infection. The secondary objective was to analyze the role of AATD and its related genetic mutations in such functional alterations.

## 2. Material and Methods

### 2.1. Study Design

This was a prospective, cohort study that included patients admitted to hospital for COVID-19 pneumonia from February to May 2020 (i.e., non-vaccinated patients from the “first wave” of COVID-19 in Spain) who were referred for follow-up to a post-COVID-19 respiratory clinic at the Virgen de las Nieves University Hospital in Granada, Spain. The patients were followed from May 2020 to May 2021. All were enrolled consecutively, and three follow-up visits were scheduled (Figure 1).

For COVID-19 diagnosis, RT-PCR from upper respiratory tract samples (nasopharyngeal or oropharyngeal swab) or lower respiratory tract samples (sputum collection) with antibody serology (IgM and IgG) by ELISA were used. The patients were followed for one year after the acute infection.

The first follow-up visit was two months after discharge from hospital. At that visit, previous medical history, characteristics of the hospital admission, baseline parameters and physical data were collected, and an X-ray was performed. The second follow-up was six months after discharge. Pulmonary function tests (PFT) and a 6-min walk test (6MWT) were performed. The last follow-up was carried out 12 months after hospital discharge and included laboratory tests and AAT levels along with genotyping and repeat PFT for those who had any abnormality in previous tests.

The study was carried out in accordance with the requirements stipulated in the Declaration of Helsinki (2013 revision) and Spanish Organic Law 15/1999 on the Protection of Personal Data. The study was approved by the Hospital’s Ethics Committee, assigning it the internal code 1017-N-20.

### 2.2. Study Population

The inclusion criteria were: patients over 18 years of age with a confirmed diagnosis of SARS-CoV-2 infection (according to international recommendations [14]), hospitalization for COVID-19 pneumonia and informed consent. Exclusion criteria were: patients without confirmation of SARS-CoV2 infection, patients with underlying immunosuppression and patients in whom therapeutic efforts were limited.

### 2.3. Study Variables

#### 2.3.1. During Hospitalization

Data were collected on demographics, medical history and characteristics of the hospitalization, especially the laboratory parameters related to the severity or mortality of the disease [15] (Appendix A).

During the patients’ hospital stay, chest X-rays in posterior–anterior and lateral projections were performed at diagnosis. Results were reported according to the recommendations of the Sociedad Española de Radiología Médica (SERAM) [Spanish Society of Medical Radiology], the international standard nomenclature defined by the Fleischner Society glossary, and available publications at the time of reporting [16,17]. The features of each X-ray performed were described in terms of density (alveolar, ground glass or mixed), distribution (central, peripheral or diffuse), location (unilateral or bilateral) and extension (unilobar or multilobar).

#### 2.3.2. During follow-up

At the 6-month follow-up, PFT with spirometric measurements, lung volumes and carbon monoxide diffusing capacity and a 6-min walk test (6MWT) were scheduled. The functional examination was carried out by experienced personnel using Jaeger MasterScreen Body whole body plethysmograph equipment (CareFusion Germany 234 GmbH, Hoechberg, Germany). Reference values and acceptability criteria were based on European and Spanish standards [18,19].

At the 12-month follow-up visit, further PFT were performed only on patients with functional alterations at the previous check-up. Plasma concentrations of AAT and C-reactive protein (CRP) were determined in all patients to rule out temporary elevation of AAT levels due to inflammation [20]. Genotyping of AAT was performed from a mouth swab using the Progenika A1AT Genotyping Test (Progenika Biopharma, a Grifols Company, Derio, Spain). The test allows simultaneous analysis of up to 384 samples per batch and is able to identify the 14 most frequent deficiency variants of the *SERPINA1* gene; details are available elsewhere [21].

The Progenika clinical service laboratory sequenced the *SERPINA1* gene if they did not find any of the 14 mutations and the serum AAT level was <60 mg/dL, or at the request of the treating physician. Sequencing of the seven exons of the gene was performed using latest generation-NGS techniques (MiSeq, Illumina Inc., San Diego, CA, USA); details are available elsewhere [21].

### 2.4. Statistical Analysis

In the descriptive analyses, continuous variables are given as mean and standard deviation. Categorical variables are given as numbers and percentages. For the comparison of quantitative data between the two groups for PFT variables altered at 12 months and abnormal AAT values, Student’s t test for parametric independent data and the Mann–Whitney U test for non-parametric data were used. For quantitative data comparisons between qualitative variables with three or more groups, ANOVA was applied with the Bonferroni correction for multiple comparisons in the case of normality and the Kruskal–Wallis test otherwise. The association between the categorical variables was assessed by means of contingency tables applying the chi-square test for individual comparisons or Fisher’s exact test for multiple comparisons. Univariate binary logistic regression analysis was used to study the association between the different variables related to baseline comorbidities, demographic, clinical and physical characteristics and laboratory tests and functional respiratory recovery at 12 months after hospital discharge. Finally, a multivariate model was adjusted from those variables that showed a significant association in the univariate analysis. Data were processed for analysis using IBM SPSS Version 25.0 (IBM Corp, Armonk, New York, NY, USA) and/or mathematical computing R software. A *p* value < 0.05 was considered statistically significant.

## 3. Results

Out of 317 patients screened, a total of 182 patients hospitalised for SARS-CoV-2 infection were referred to post-COVID-19 respiratory clinics for review. The follow-up study lasted for one year after the acute infection with 168 patients (92.3%) attending the 6-month check-up and 157 (86.3%) the 12-month check-up (Figure 1).

The clinical characteristics of the study population separated according to normal or impaired lung function at 12 months are shown in Table 1. Details of laboratory parameters are provided in Appendix A.

The study population was predominantly males (93 [59.2%]) with a mean age of 59.9 years (±12.9) and a mean BMI of 29.4 (±5.0). Ninety-eight patients (62.4%) were active smokers or former smokers with a mean cumulative smoking burden index (ICAT) of 12.4 (±21.6) pack-years. A total of 68 patients (43.3%) had two or more baseline comorbidities with a Charlson index of 3 or higher in 64 patients (40.8%). The mean hospital stay was 11.5 days (±2.8), and the most common clinical manifestations while in hospital (affecting over 25% of patients) were fever, dyspnoea, cough, fatigue, musculoskeletal pain and gastrointestinal symptoms (Table 1). The admission chest X-ray showed pneumonia in 142 cases (90.4%). Sixty-seven patients (43.2%) received corticosteroids (prednisone and/or methylprednisolone) and 135 (87.1%) received antiretrovirals during treatment.

### 3.1. Association between the Patients Clinical Characteristics and Lung Function 12 Months after Hospital Discharge

Of the total of 157 patients who completed the study, 49 (31.2%) met the criteria for abnormal PFT at 12 months, i.e., they had impaired lung function. The remaining 108 patients (68.8%) had PFT within normal limits.

The comparison of the two groups (abnormal vs. normal PFT) using univariate binary logistic regression analysis is shown in Table 2.

After univariate analysis of the association of the different variables with the risk of having impaired lung function, a multivariate model was adjusted, which initially contained all the variables. From this model, a predictive model was adjusted that included the following as predictors of alterations of lung function 12 months after hospitalization: the Charlson index; pneumonia with a mixed and/or central distribution; anaemia on admission; admission to the ICU; need for treatment with corticosteroid boluses; persistence of abnormal respiratory sounds at the 6-month check-up; and lactate dehydrogenase (LDH) elevation at the 12-month follow-up (Table 2).

### 3.2. Lung Function at 6 and 12 Months after Hospitalization

All patients who attended the 6-month follow-up were asked to undergo PFT and a 6MWT, but only 150 completed these tests. Of that 150, 67 (42.7%) had some type of functional impairment. The same patients were asked to undergo repeat PFT at the 12-month check-up, and 49 patients (31.2%) continued to have impaired lung function. Therefore, 18 patients (11.5%) with abnormal PFT at 6 months had recovered functionally at 12 months (Figure 2). The serial values for the PFT at 6 and 12 months post discharge in the patients who had pulmonary functional alterations are shown in Table 3.

### 3.3. Association between AAT Levels and Genotyping and Functional Recovery

The data from the 157 patients who attended the third follow-up were categorised according to genotype with the corresponding plasma levels of AAT and elevated CRP value > 5 mg/dL (Appendix A); 83.4% of hospitalised patients had normal AAT genotype (Pi*MM), with plasma AAT levels of 128.6 ± 1.4 mg/dL. Other genotypic variants found were Pi*MS (14.6%), Pi*MZ (1.3%) and Pi*M/P Lowell (0.6%) with plasma AAT levels of 116.4 ± 3.2, 89.0 ± 15.0 and 114.8 ± 0.0 mg/dL, respectively.

Figure 3 shows a graph of the mean plasma levels of AAT for each category of AAT genotype analysed with patients stratified based on the CRP level (categorised as normal if ≤5 mg/L or elevated if >5 mg/L) for all hospitalised patients and for the subgroups with normal or impaired lung function at 12 months.

Table 4 shows how normal AAT plasma levels (range 83–220 mg/dL) or abnormal levels, along with the allelic genotype detected (Pi*MM, Pi*MS, Pi*MZ and Pi*M/P Lowell), influence the recovery of respiratory function variables at 6 and 12 months of follow-up.

At 6 months, the functional parameters of forced expiratory volume in the first second (FEV1) (69.9 ± 22.8 vs. 99.9 ± 17.8; *p* = 0.02), FEV1/forced vital capacity (FVC) (66.4 ± 13.6 vs. 78.4 ± 7.0, *p* = 0.018) and total lung capacity (TLC) (79.5 ± 19.1 vs. 102.9 ± 14.8, *p* = 0.029) were significantly lower in the abnormal AAT group than in the normal AAT group. However, at 12 months, only TLC values (75.0 ± 12.2 vs. 104.9 ± 20.1, *p* = 0.041) showed a significant decrease in the abnormal AAT group.

For allele genotyping and respiratory function variables at 6 months, significant differences were found in the carbon monoxide transfer by a single breath (DLCO) (*p* = 0.045) and the distance covered in the 6MWT (*p* = 0.040). The analysis of multiple comparisons (Appendix A) showed differences between the groups with the Pi*MM and Pi*MZ genotypes for TLC (mean difference 28.7 ± 10.6; *p* = 0.023), DLCO (mean difference 41.96 ± 15.5; *p* = 0.024) and the distance covered in the 6MWT (mean difference 210.2 ± 74.9 m, *p* = 0.017).

## 4. Discussion

Published reports issued in the weeks or months after discharge from hospital for COVID-19 pneumonia describe patients with varying degrees of persistent symptoms and radiological and functional abnormalities [22,23]. In the present study, risk factors associated with impaired lung function in patients who were hospitalised for COVID-19 were identified: a higher Charlson index, severe pneumonia with anemia, need for corticosteroid boluses of methylprednisolone, admission to the ICU. Our results are in line with those published to date on risk factors associated with a worse clinical course, acute respiratory distress syndrome (ARDS) and death during hospitalization [24,25]. In this study, high levels of haemoglobin, lymphocytes, ferritin, albumin and troponin significantly reduced the risk of impaired lung function. The elevation of troponin and ferritin has previously been reported as a risk factor for poor clinical outcome, in contrast to our results. One explanation for their protective role may be the fact that these parameters increase early in cases with a poor clinical course and development of the inflammatory cascade thus potentially allowing early diagnosis and management.

We also found that close to one third of the patients (n = 49; 31.2%) continued to have impaired lung function at 12 months. The most affected functional parameters being FEV1 and DLCO. To date, there are studies that report on short- and medium-term functional outcomes [22,26]. However, studies evaluating one-year results have only recently become available. Wu et al. described persistent physiological (DLCO: 88% reduction of predicted) and radiographic (24% of patients) abnormalities in some patients 12 months after discharge for COVID-19 treatment [26]. Liu et al. reported physiological, laboratory, radiological or electrocardiogram abnormalities, with those related to renal, cardiovascular and liver function being particularly common, in patients who recovered from COVID-19 up to 12 months post-discharge [27]. Huang et al. reported 30% of patients with dyspnoea and 26% of patients with anxiety or depression at the 12-month visit among COVID-19 survivors [28]. In general for our study, all the functional variables improved from 6 to 12 months. However, only TLC and the distance covered in metres in the 6MWT reached statistical significance. Both variables showed a statistically significant decrease despite improvement in lung volumes. This may be due to persistent muscle weakness, which can cause dyspnoea on exertion and limit performance in this test.

With regard to AAT levels, two patients were in the abnormal range (both below normal), one with normal and one with impaired lung function. The mean level of AAT was similar in patients with normal and impaired lung function (124.4 ± 15.5 and 129.1 ± 18.2, respectively). However, there were 28 patients with normal levels of AAT and CRP > 5 mg/L. It is worth noting that AAT is an acute-phase reactant and its plasma levels have been shown to increase 2–3 times in response to inflammatory or infectious stimuli, similar to CRP [20]. Complete genome sequencing was requested for these patients, but there were no pathogenic variants found.

The allelic variants of AAT in our patients were distributed in a similar way to the general population [29]; Pi*MM being predominant (131 patients: 83.4%) and deficient genotypes being far less common (Pi*MS 22 patients: 14.6%, Pi*MZ 2 patients: 1.3% and Pi*M/P Lowell 1 patient: 0.6%). Significant differences between the groups with normal and impaired lung function at 12 months were seen only in the normal genotype (Pi*MM). However, it should be noted that the two Pi*MZ cases and half of the Pi*MS cases were in the group of patients with impaired lung function. Moreover, the analysis of multiple comparisons revealed that there were statistically significant differences between PiMM and PiMZ genotypes in the patients with normal pulmonary function regarding DLCO and TM6M. In other study, the most common mild AATD genotypes were associated neither with increased SARS-CoV-2 infection rates nor with increased SARS-CoV-2 fatalities. The numbers of patients with severe AATD cases were too low to allow definitive conclusions [30].

With this study being single-centre and observational, there is a potential location bias. However, our study does have several strengths. It has a large sample size and included patients with moderate or severe COVID-19 with previous hospitalization who continue to have long-term functional sequelae. Many characteristics that influenced their recovery had not previously been reported.

In conclusion, our study found that close to one third of COVID-19 patients still showed impaired lung function 12 months after infection. The presence of a high Charlson index, severe pneumonia with anaemia, need for corticosteroid boluses, admission to the ICU and low AAT levels or Pi*MZ deficiency allele variants predisposed patients to impaired lung function at 12 months after hospitalisation due to COVID-19.

## Figures and Tables

**Figure 1 healthcare-10-02341-f001:**
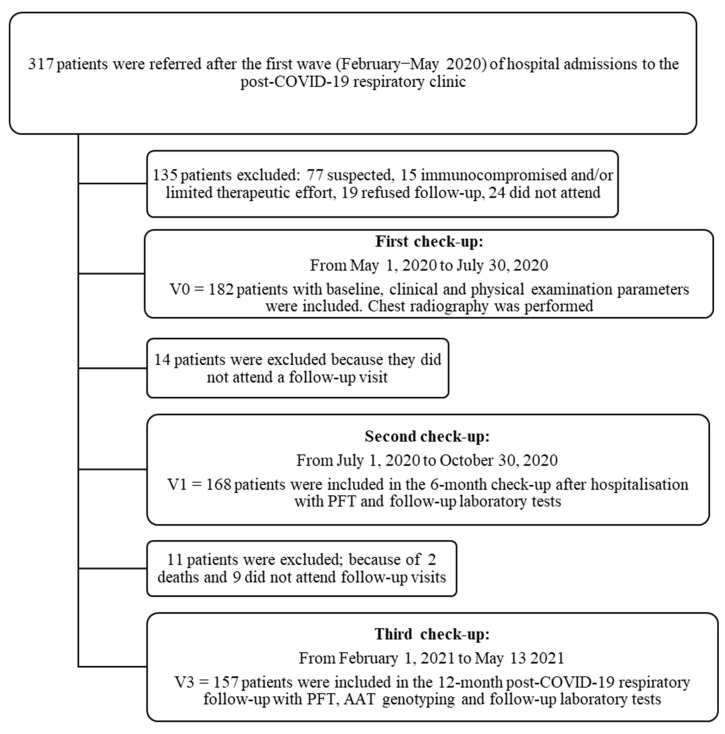
Patient flow chart after acute COVID-19 respiratory infection with first, second and third follow-up visits carried out at the University Hospital Virgen de las Nieves.

**Figure 2 healthcare-10-02341-f002:**
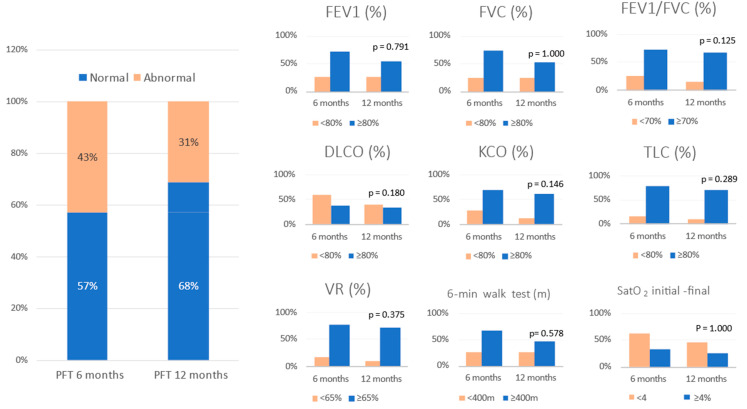
Evolution of pulmonary function tests at 6 and 12 months follow-up. Data are expressed as percentege (%). Abbreviations: FVC = forced vital capacity; FEV1 = forced expiratory volume in the first second; TLC = total lung capacity; RV = residual volume; DLCO = carbon monoxide transfer by single breath; KCO= diffusion constant for carbon monoxide; VR = residual volume; 6MWT = six-minute walking test; SatO_2_ = oxygen saturation.

**Figure 3 healthcare-10-02341-f003:**
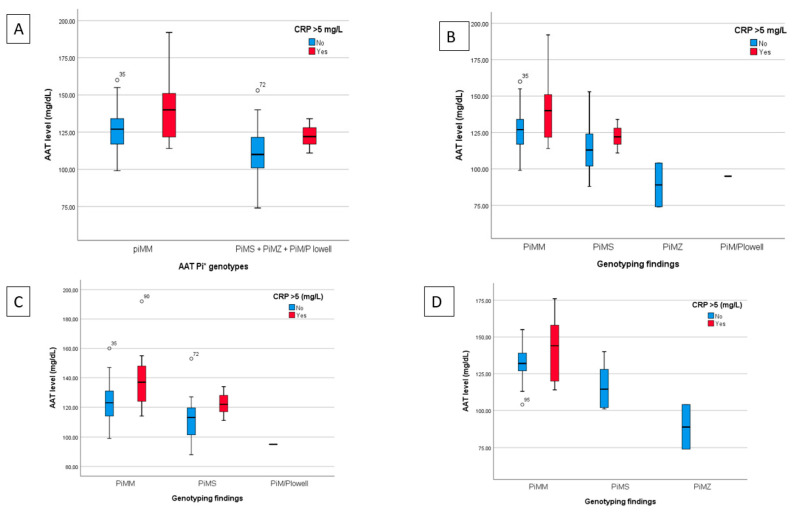
Findings from the plasma determination of AAT, allelic genotype and CRP determination for all hospitalised patients and for the subgroups of patients with normal or impaired lung function at 12 months. (**A**) Breakdown of AAT levels classified by normal genotype (Pi*MM) or unified heterozygous deficiencies (Pi*MS, Pi*MZ and Pi*M/P Lowell) and stratified by normal or elevated CRP for all hospitalised patients. (**B**) Comparison of AAT levels based on genotyping (Pi*MM, Pi*MS, Pi*MZ and Pi*M/P Lowell) and the normal or high CRP value for hospitalised patients. C and D) AAT values analyzed based on genotype and normal or elevated CRP of the subgroups of patients with normal (**C**) or impaired lung function (**D**) at the 12 month follow-up.

**Table 1 healthcare-10-02341-t001:** Baseline comorbidities, demographics, clinical and physical characteristics of the patients at admission, during hospitalization and at the 12-month follow-up visit.

Baseline Characteristics	Lung Function at 12 Months [Mean ± SD or n (%)]		*p*-Value
	Normal (n = 108)	Impaired (n = 49)	
Age	57.4 ± 12.5	62.4 ± 13.0	0.026
Male	68 (63)	25 (51)	0.272
Smoking history:			0.201
Active smoker	6 (5.6)	3 (6.1)	
Never smoked	64 (59.3)	25 (51)	
Former smoker	38 (35.2)	21 (35.6)	
Cumulative smoking burden index (smokers and former smokers)	11.3 ± 19.9	14.8 ± 24.9	0.039
BMI (kg/m^2^)	29.4 ± 5.0	29.1 ± 5.0	0.938
**Previous medical history**			
≥2 baseline comorbidities	37 (34.3)	31 (63.3)	0.006
HBP	45 (41.7)	25 (51)	0.413
DM2	10 (9.3)	13 (26.5)	0.006
Dyslipidemia	23 (21.5)	14 (28.6)	0.599
Coronary heart disease	3 (2.8)	1 (2)	0.075
Cardiac arrhythmias	4 (3.7)	4 (8.2)	0.715
Cerebrovascular disease	2 (1.9)	3 (6.1)	0.391
Baseline respiratory comorbidities	22 (20.4)	20 (40.8)	0.008
COPD	3 (2.8)	8 (16.3)	0.006
Baseline liver disease	3 (2.8)	2 (4.1)	0.638
Chronic kidney disease	6 (5.6)	4 (8.2)	0.633
Active cancer	1 (2%)	1 (0.6%)	0.401
Previous cancer history < 5 years	5 (4.7)	5 (10.2)	0.526
Charlson index ≥ 3	32 (29.6)	32 (65.3)	0.002
**Signs and symptoms on admission**			
Fever (>37.5 °C)	103 (72.5)	39 (79.6)	0.002
Dyspnoea	62 (57.9)	39 (79.6)	0.010
Cough	79 (73.8)	38 (77.6)	0.638
Exhaustion	83 (77.6)	31 (63.3)	0.064
Musculoskeletal pain	60 (56.1)	17 (35.4)	0.019
Anosmia/ageusia	24 (22.4)	13 (26.5)	0.449
Gastrointestinal symptoms	27 (64.3)	15 (30.6)	0.441
Cardiac symptoms	4 (3.7)	3 (6.1)	0.441
Neurological symptoms	16 (15)	8 (16.3)	0.754
Dermatological symptoms	1 (0.9)	1 (2.0)	0.270
ARDS (PaO_2_/FiO_2_ < 300 mmHg)	21 (19.6)	19 (38.8)	0.012
**Hospitalization**			
Mean length of stay (days)	9.9 ± 11.2	14.8 ± 14.0	0.043
Pneumonia on admission	101 (93.5)	41 (83.7)	0.047
Characteristics of pneumonia on chest X-ray (n = 142)			
-Density:			
Alveolar	25 (24.8)	13 (31.7)	0.247
Ground-glass opacity (GGO)	19 (18.8)	5 (12.2)	
Mixed	57 (56.4)	23 (56.1)	
-Distribution:			
Central	6 (5.9)	8 (19.5)	0.047
Peripheral	68 (67.3)	17 (41.5)	
Mixed	27 (26.7)	16 (39)	
-Location:			
Unilateral	10 (9.9)	5 (12.2)	0.505
Bilateral	91 (90.1)	36 (87.8)	
-Involvement:			
Single lobe	9 (8.9)	3 (7.3)	0.827
Multilobar	92 (91.1)	38 (92.7)	
Admission to the ICU	3 (2.8)	6 (12.2)	0.030
Endotracheal intubation (ETI)	2 (1.9)	6 (12.2)	0.017
Treatment received:			
Corticosteroids (prednisone, methyprednisolone)	39 (58.2)	28 (41.8)	0.019
Antivirals	92 (68.1)	43 (31.9)	0.625
Tocilizumab	7 (43.8)	9 (56.3)	0.031
Anakinra	3 (75)	1 (25)	0.679
Antimalarials (hydroxychloroquine)	102 (69.4)	45 (30.6)	0.182

The results are expressed as frequency (%) and mean (±SD). Abbreviations: BMI = body mass index, ARDS = acute respiratory distress syndrome, IL-6 = interleukin 6; COPD = chronic obstructive pulmonary disease; DM2 = diabetes mellitus-2; HBP = high blood pressure; ICU = intensive care unit; ETI = endotracheal intubation.

**Table 2 healthcare-10-02341-t002:** Risk factors associated with impaired lung function at 12 months for the bivariate and multivariate analysis.

Covariate	OR ^a^	*p*-Value	95% CI for OR	Explained Variance, R^2^
**(A) Univariate logistic regression analysis**				
Age	1.032	0.026	1.004–1.061	0.046
Charlson index	1.421	<0.001	1.186–1.703	0.139
≥2 comorbidities	3.305	0.001	1.635–6.680	0.100
DM2	3.539	0.006	1.426–8.779	0.065
Heart disease	2.833	0.075	0.899–8.931	0.028
Lung disease	2.696	0.008	1.290–5.636	0.061
COPD	6.829	0.006	1.727–27.013	0.075
Signs and symptoms on admission				
Fever	0.151	0.002	0.045–0.511	0.091
Dyspnoea	2.831	0.010	1.280–6.260	0.064
Asthenia	0.498	0.064	0.238–1.041	0.030
Myalgia	0.430	0.019	0.212–0.869	0.051
X-ray distribution				0.095
Peripheral	Ref			
Central	5.333	0.006	1.632–17.434	
Mixed	2.370	0.038	1.049–5.357	
Haemoglobin (g/dL)	0.780	0.017	0.636–0.956	0.054
Leucocytes (count ×10^3^/µL)	1.172	0.011	1.037–1.324	0.061
Neutrophils (count ×10^3^/µL)	1.175	0.024	1.022–1.352	0.051
Lymphocytes (%)	0.957	0.044	0.916–0.999	0.041
NLR	1.144	0.006	1.040–1.259	0.080
Platelets (count ×10^3^/µL)	1.004	0.012	1.001–1.008	0.061
Hospitalization				
ICU admission	4.884	0.030	1.168–20.420	0.045
ETI	7.395	0.017	1.436–38.089	0.060
ARDS	2.594	0.012	1.229–5.474	0.055
Corticosteroid therapy	2.291	0.019	1.149–4.566	0.050
Corticosteroid boluses	3.665	<0.001	1.765–7.612	0.107
Tocilizumab	3.182	0.031	1.109–9.128	0.041
Haemoglobin (g/dL)	0.759	0.008	0.620–0.930	0.076
Albumin (g/dL)	0.439	0.048	0.195–0.991	0.043
LDH (U/L)	1.004	0.011	1.001–1.007	0.065
Procalcitonin	10.160	0.052	0.982–105.105	0.062
IL-6 (pg/mL)	1.107	0.090	0.997–1.036	0.120
LTOT	4.921	0.003	1.701–14.235	0.080
Length of hospital stay (days)	1.003	0.043	1.001–1.066	0.045
Follow-up				
Haemoglobin (g/dL)	0.763	0.012	0.619–0.941	0.061
Platelets (count ×10^3^/µL)	1.005	0.028	1.001–1.010	0.051
LDH (U/L)	1.011	0.017	1.002–1.021	0.063
Ferritin (ng/mL)	0.995	0.023	0.991–0.999	0.065
Troponin (ng/L)	0.782	0.007	0.654–0.934	0.100
Vitamin D (ng/mL)	0.955	0.058	0.910–1.002	0.042
Abnormal respiratory sounds	7.186	0.018	1.395–37.023	0.059
**(B) Multivariate logistic regression analysis**				
Charlson index	1.336	0.030	1.029–1.735	0.534
X-ray distribution				
Peripheral	Ref			
Central	10.820	0.004	2.093–55.934	
Mixed	4.855	0.014	1.374–17.154	
Admission haemoglobin (g/dL)	0.604	0.006	0.422–0.864	
ICU admission	33.184	0.012	2.180–505.072	
Methylprednisolone boluses	3.447	0.043	1.039–11.433	
Follow-up LDH (U/L)	1.025	0.004	1.008–1.042	
Abnormal respiratory sounds	15.157	0.027	1.011–227.244	

OR = odds ratio; Ref = reference category; LDH = lactate dehydrogenase; NRL = neutrophil/lymphocyte ratio; IL-6 = interleukin 6; LTOT = long-term oxygen therapy. ICU = intensive care unit; ETI = endotracheal intubation; ARDS = acute respiratory distress syndrome; DM2 = Diabetes mellitus type-2. (^a^) The OR account for a 1 unit increase in each of the independent variables.

**Table 3 healthcare-10-02341-t003:** Serial values of pulmonary function tests at 6 and 12 months in the subgroup of patients with impaired lung function. Data are expressed as percentage (%) or mean ± standard deviation (SD).

Respiratory Function Parameters	At 6 Month (n = 67) (Mean ± SD)	At 12 Month (n = 49) (Mean ± SD)	Difference (Mean ± SD)	95% CI	*p*-Value
FVC (%)	91.2 ± 17.5	92.2 ± 19.7	−0.98 ± 12.4	−4.340–2.369	0.558
FEV1 (%)	87.1 ± 20.3	90.8 ± 21.6	−3.6 ± 16.0	−7.958–0.708	0.099
FEV1/FVC	75.0 ± 11.5	77.4 ± 10.2	−2.4 ± 9.4	−4.909–0.176	0.067
TLC (%)	98.0 ± 18.4	102.6 ± 21.9	−4.6 ± 14.6	−8.671–0.555	0.027
DLCO (%)	73.6 ± 19.0	76.6 ± 13.5	−2.9 ± 17.6	−7.925–2.069	0.245
KCO (%)	92.3 ± 20.8	95.1 ± 16.1	−2.8 ± 17.9	−7.866–2.282	0.274
RV (%)	100.4 ± 30.8	105.6 ± 47.0	−5.3 ± 38.4	−15.96–5.41	0.327
Distance 6MWT (m)	466.4 ± 107.9	430.0 ± 114.7	36.4 ± 79.9	12.95–59.89	0.003
Initial SaO_2_ 6MWT %)	96.5 ± 1.8	95.8 ± 2.4	0.71 ± 2.6	−0.040–1.456	0.063
Final SaO_2_ 6MWT %)	93.7 ± 4.0	92.7 ± 4.2	1.0 ± 3.3	0.000–1.917	0.05
Initial—final SaO_2_ 6MWT (%)	2.9 ± 4.2	3.2 ± 4.3	−0.25 ± 4.1	−1.452–0.952	0.677

FVC = forced vital capacity; FEV1 = forced expiratory volume in the first second; TLC = total lung capacity; RV = residual volume; DLCO = diffusing capacity for carbon monoxide; 6MWT = six minute walk test; SaO_2_ = oxygen saturation.

**Table 4 healthcare-10-02341-t004:** Association between alpha-1 antitrypsin (AAT) levels results and allelic genotype with pulmonary function test parameters at 6 and 12 month follow-up visits.

	AAT Levels (mg/dL)			Genotype				
	Normal	Abnormal	*p*-Value	Pi*MM	Pi*MS	Pi*MZ	Pi*MP Lowell	*p*-Value
**(A) Abnormal PFT (mean ± SD) at 6 months (N = 66)**								
FEV1 %	99.9 ± 17.8	69.9 ± 22.8	0.020	100.4 ± 18.7	92.1 ± 17.2	78.9 ± 35.4	114.8	0.086
FVC %	106.7 ± 76.0	79.7 ± 10.9	0.618	101.0 ± 16.4	134.0 ± 183.9	86.3 ± 20.2	113.7	0.244
FEV1/FVC %	78.4 ± 7.0	66.4 ± 13.6	0.018	78.0 ± 8.1	77.0 ± 9.1	69.8 ± 18.5	79.1	0.549
TLC %	102.9 ± 14.8	79.5 ± 19.1	0.029	103.5 ± 14.7	102.6 ± 16.3	74.9 ± 12.5	107.8	0.065
DLCO %	88.6 ± 22.1	80.4 ± 40.7	0.607	89.5 ± 22.2	85.2 ± 19.8	47.7 ± 5.6	103.4	0.045
KCO %	100.2 ± 17.7	110.0 ± 25.5	0.438	99.9 ± 18.0	100.1 ± 17.4	79.4 ± 17.8	109.4	0.414
RV %	102.1 ± 26.4	76.8 ± 22.9	0.181	100.4 ± 24.8	114.2 ± 36.1	69.5 ± 12.6	93.1	0.055
6MWT Distance (m)	507.0 ± 110.3	568.5 ± 115.3	0.436	516.0 ± 100.8	494.5 ± 115.1	306.0 ± 256.0	-	0.040
6MWT Initial SaO_2_ (%)	96.5 ± 1.5	95.5 ± 2.1	0.346	96.6 ± 1.6	96.6 ± 1.5	95.0 ± 2.8	-	0.567
6MWT Final SaO_2_ (%)	95.0 ± 3.8	91.5 ± 5.0	0.198	95.0 ± 3.9	95.0 ± 3.9	89.5 ± 2.1	-	0.224
**(B) Abnormal PFT (mean ± SD) at 12 months (N = 49)**								
FEV1 %	95.9 ± 20.4	69.4 ± 23.6	0.075	97.0 ± 19.7	94.1 ± 24.1	64.1 ± 16.6	-	0.083
FVC %	77.7 ± 9.2	64.9 ± 14.4	0.257	98.3 ± 19.5	95.6 ± 19.8	71.1 ± 2.8	-	0.149
FEV1/FVC %	96.9 ± 2.3	81.3 ± 8.2	0.059	77.6 ± 8.9	76.7 ± 13.0	70.8 ± 22.8	-	0.616
TLC %, SD	104.9 ± 20.1	75.0 ± 12.2	0.041	105.7 ± 20.5	100.6 ± 16.7	74.9 ± 12.2	-	0.087
DLCO %, SD	82.7 ± 15.1	75.5 ± 40.2	0.525	82.8 ± 15.5	81.1 ± 11.0	47.0	-	0.067
KCO %, SD	111.9 ± 118.7	101.6 ± 23.5	0.903	111.3 ± 117.1	98.0 ± 15.5	84.9	-	0.916
RV %, SD	108.7 ± 41.8	59.9 ± 5.9	0.105	109.8 ± 42.3	92.2 ± 4	66.5 ± 15.3	-	0.196
6MWT Distance (m)	442.8 ± 104.2	567.5 ± 24.7	0.098	443.3 ± 102.0	441.9 ± 107.0	550.0	-	0.588
6MWT Initial SaO_2_ (%)	95.7 ± 2.2	97.5 ± 2.1	0.258	95.7 ± 2.3	95.5 ± 1.9	99.0	-	0.329
6MWT Final SaO_2_ (%)	93.3 ± 3.9	94.5 ± 2.1	0.662	92.9 ± 4.0	94.6 ± 2.7	96.0	-	0.317

Abbreviations: SD = standard deviation FVC = forced vital capacity; FEV1 = forced expiratory volume in the first second; TLC = total lung capacity; RV = residual volume; DLCO = diffusing capacity for carbon monoxide transfer in single breath; 6MWT = six minute walk test; SaO_2_ = oxygen saturation; AAT= Alpha-1 antitrypsin; PFT= pulmonary function test.

## Data Availability

The datasets used and/or analysed during the current study are available from the corresponding author on reasonable request.

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
