# Peer review of "Abnormal Alpha-1 Antitrypsin Levels and Other Risk Factors Associated with Lung Function Impairment at 6 and 12 Months after Hospitalization Due to COVID-19: A Cohort Study"

_healthcare, 2022, doi:10.3390/healthcare10122341_

Round 1
Reviewer 1 Report
Correct the expression "DAAT" used in the last sentence of the introduction to be "AATD".
Correct the expression "prospective, observational, cohort study" used in the first sentence of the Study design section to be "prospective, cohort study".
In the statistical analysis section, "Analysis of variance (ANOVA) or the Kruskal-Wallis test was used to study the association between quantitative and qualitative variables with more than two categories, applying the Bonferroni correction for multiple comparisons. You used the phrase ''. This sentence needs to be revised. Because these two tests are only used to compare quantitative variables.
Please write the sentence "In the descriptive analyzes, measures of central tendency (mean) and dispersion (standard deviation) were used for continuous variables and the distribution of absolute and relative frequencies for categorical variables" in a simpler and more understandable way. For example, you can write: continuous variables are given as mean and standard deviation. Categorical variables are given as numbers and percentages.
It is seen that many variables given in Table S2 are not normally distributed. Please check all tables (main and supplemantary) and add a column to the end of the table and give p-values in that columns. Please delete the columns titled "Total" in the all tables.
In short, the statistical analysis of this study should be done by an expert statistician. Both the methodology section and the results section should be revised.
Author Response
REVIEWER 1
We thank the reviewer for his/her helpful comments which greatly contributed to improve our manuscript.
Point 1: Correct the expression "DAAT" used in the last sentence of the introduction to be "AATD"
Response 1: The misspelling has been corrected following the indication of the reviewer (page 4, second paragraph).
Point 2: Correct the expression "prospective, observational, cohort study" used in the first sentence of the Study design section to be "prospective, cohort study".
Response 2: The sentence has been corrected following the advice of the reviewer (page 5, first paragraph).
- Point 3: In the statistical analysis section, "Analysis of variance (ANOVA) or the Kruskal-Wallis test was used to study the association between quantitative and qualitative variables with more than two categories, applying the Bonferroni correction for multiple comparisons. You used the phrase ''. This sentence needs to be revised. Because these two tests are only used to compare quantitative variables.
Response 3: The reviewer is correct. In the revised manuscript the text has been reworded to clarify that for quantitative data comparisons between qualitative variables with three or more groups, the ANOVA test was applied with the Bonferroni correction for multiple comparisons in the case of normality, and the Kruskal-Wallis test otherwise (page 9, first paragraph).
Point 4: Please write the sentence "In the descriptive analyzes, measures of central tendency (mean) and dispersion (standard deviation) were used for continuous variables and the distribution of absolute and relative frequencies for categorical variables" in a simpler and more understandable way. For example, you can write: continuous variables are given as mean and standard deviation. Categorical variables are given as numbers and percentages.
Response 4: The sentence has been modified following the reviewer’s advice (page 9, first paragraph).
Point 5: It is seen that many variables given in Table S2 are not normally distributed. Please check all tables (main and supplemantary) and add a column to the end of the table and give p-values in that columns. Please delete the columns titled "Total" in the all tables.
Response 5: All the tables have been checked thoroughly. Hence, in the revised manuscript Tables 1, S2 and S3 have been modified following the indications of the reviewer (addition of a column with p-values, and deletion of the “Total” column).
Point 6: In short, the statistical analysis of this study should be done by an expert statistician. Both the methodology section and the results section should be revised.
Response 6: Both the methodology section and the results section have been revised accordingly. We hope that the changes performed have addressed the reviewer’s concerns.
Reviewer 2 Report
Comments to Author:
I think the aim of the study is very important since there are many patients who seem to suffer from post COVID-19 infection syndrome. To make the report more useful for readers, the authors are kindly asked to check the points below.
1. How were the patients diagnosed as having COVID-19 (PCR, antigen, clinically…)?
2. Were vaccinated patients included? (either before diagnosis of COVID-19 or during observation periods)?
3. What was the kind of corticosteroid in this study (dexamethasone, prednisone…)?
Author Response
REVIEWER 2
We thank the reviewer for his/her comments which greatly contributed to improve our manuscript.
Point 1: How were the patients diagnosed as having COVID-19 (PCR, antigen, clinically…)?
Response 1: For COVID-19 diagnosis, the RT-PCR from upper respiratory tract samples (nasopharyngeal or oropharyngeal swab) or lower respiratory tract samples (sputum collection) with antibody serology (IgM and IgG) by ELISA, were used. This information has been included in the revised manuscript (Study design: page 5, second paragraph).
Point 2: Were vaccinated patients included? (either before diagnosis of COVID-19 or during observation periods)?
Response 2: All patients were non-vaccinated because they were recruited from the “first wave” of COVID-19 in Spain (February to May 2020), before vaccines were available. This information has been included in the revised manuscript (Study design: page 5, first paragraph).
Point 3: What was the kind of corticosteroid in this study (dexamethasone, prednisone…)?
Response 3: The kind of corticoid has been added in Tables 1 (prescribed during hospitalization) and 2 (multivariate analysis), as well as in the main text (Results: page 15, first paragraph; Discussion: page 27, first paragraph).
Round 2
Reviewer 1 Report
Thank you
Author Response
The only comment of the reviewer that we can see is "Thank you". Therefore we understand that the reviewer is satisfied with our responses to their concerns.